# VARIATIONAL INFERENCE WITH SINGULARITY-FREE PLANAR FLOWS

## ABSTRACT

Variational inference is a method for approximating probability distributions. The approximation quality depends on the expressiveness of variational distributions. Normalizing flows provide a way to construct a flexible and rich family of distributions. Planar flow, an early studied normalizing flow, is simple but powerful. Our research reveals a crucial insight into planar flow's constrained parameters: they exhibit a non-removable singularity in their original reparameterization. The gradients of the associated parameters diverge to infinity in different directions as they approach to the singularity, which creates a potential for the model to overshoot and get stuck in some undesirable states. We then propose a new reparameterization to eliminate the singularity. The resulting singularity-free planar flows are more stable in training and demonstrate better performance in variational inference tasks.

## 1 INTRODUCTION

Variational inference is a method for approximating probability distributions, in particular, posterior distributions of latent variables and parameters in Bayesian models (Jordan et al., 1999; Wainwright & Jordan, 2008; Blei et al., 2017). It seeks the best distribution within a given family of distributions by optimization. The use of stochastic optimization has enabled variational inference to handle massive data sets efficiently (Paisley et al., 2012; Hoffman et al., 2013).

The most common optimization criterion used in variational inference is the Kullback-Leibler (KL) divergence. Let $p^*(\mathbf{z})$ be the target distribution and $\mathcal{Q}$ be the family of approximate distributions. The optimal approximate distribution is defined as $q^*(\mathbf{z}) = \arg\min_{q(\mathbf{z}) \in \mathcal{Q}} D_{\mathrm{KL}}[q(\mathbf{z}) \| p^*(\mathbf{z})]$. Often the target distribution is a posterior distribution and only known up to a multiplicative normalizing constant, $p^*(\mathbf{z}) = p(\mathbf{z}|\mathbf{x}) = p(\mathbf{z}, \mathbf{x})/p(\mathbf{x})$, where $p(\mathbf{x})$ is infeasible or requires exponential time to compute. However, it does not hinder us in solving the optimization problem, since $p(\mathbf{x})$ is an immaterial constant with respect to the optimization criterion, i.e.,

$$D_{\mathrm{KL}}[q(\mathbf{z}) \| p^*(\mathbf{z})] = \underbrace{\mathbb{E}_{q(\mathbf{z})}[\log q(\mathbf{z}) - \log p(\mathbf{z}, \mathbf{x})]}_{-\mathrm{ELBO}} + \log p(\mathbf{x}).$$

Hence, one can minimize the negative evidence lower bound ($-$ELBO) to obtain the optimal approximate distribution, $q^*(\mathbf{z})$.

As an alternative to Markov chain Monte Carlo sampling methods, variational inference is faster but only provides an approximation to a limited extent. The approximation quality depends on the choice of $\mathcal{Q}$. A desired family should be rich enough such that it includes a distribution close to the target distribution while maintaining the tractability. In practice, approximation accuracy is traded off for efficient optimization, for example, mean-field approximation (Parisi, 1988) and the Gaussian distribution approximation. These methods are efficient but often lack of approximation accuracy.

Normalizing flows provide a way to construct a flexible and rich family of distributions. Rezende & Mohamed (2015) introduced normalizing flows in the context of variational inference to improve the performance of deep latent Gaussian models. The class of flows they focused on, planar flow, has a simple structure: the only hyper-parameter is the number of layers, which controls the approximation level. It was found that planar flows can be hard to train and many layers are required to achieve a good performance. To facilitate more flexible approximations and scalability to high dimensions,

many flows are developed (e.g., Kingma et al., 2016; van den Berg et al., 2018). These extensions have demonstrated superior approximation capabilities. Nevertheless, these more intricate flow models pose challenges in terms of tuning due to the additional hyper-parameters introduced.

In this paper, we revisit the use of planar flows in variational inference and uncover a significant issue: a non-removable singularity presents in the original reparameterization of the constrained parameters. Its presence results in less stable training dynamics, often leading to suboptimal approximation performance. To address this challenge, we propose a novel reparameterization approach that effectively eliminates this singularity. Our approach enhances the stability of model training, leading to improved convergence and consequently, higher-quality approximations in variational inference. We empirically evaluate our novel method across various variational inference tasks, and the experimental results clearly demonstrate the superiority of our singularity-free planar flows.

## 2 NORMALIZING FLOWS

A comprehensive review on normalizing flows was given by Kobyzev et al. (2021). At the same time, Papamakarios et al. (2021) presented an excellent review from a unified perspective that is more tutorial in nature. Here, we provide a brief background and set up the notation.

A normalizing flow is an invertible and differentiable transformation of a random vector. While the distribution of the initial random vector is typically chosen to be some simple distribution (e.g., a Gaussian distribution), the resulting random vector could have a complex distribution after a sequence of simple transformations.

Let $\mathbf{u}$ be a $D$-dimensional random vector. Suppose that $\mathbf{u} \sim q_{\mathbf{u}}(\mathbf{u})$, where $q_{\mathbf{u}}(\mathbf{u})$ is referred to as the base distribution. Let $T$ be an invertible and differentiable transformation in $\mathbb{R}^D$. Then a normalizing flow model is given by $\mathbf{z} = T(\mathbf{u})$. The transformation $T$ is often composed by a sequence of simple functions of the same type. Suppose that

$$T = T_K \circ \cdots \circ T_1,$$
$$\mathbf{z}_k = T_k(\mathbf{z}_{k-1}), \ k = 1, \ldots, K,$$

where $\mathbf{z}_0 = \mathbf{u}$, $\mathbf{z}_K = \mathbf{z}$, and $K$ is the number of layers used in the flow model. Then the distribution of $\mathbf{z}$ can be computed by using the change of variables formula,

$$q_{\mathbf{z}}(\mathbf{z}) = q_{\mathbf{u}}(\mathbf{u}) \, | \det J_T(\mathbf{u})|^{-1}$$
$$= q_{\mathbf{u}}(\mathbf{u}) \prod_{k=1}^{K} | \det J_{T_k}(\mathbf{z}_{k-1})|^{-1}.$$

Without loss of generality, we can assume that the parameters of the base distribution, $q_{\mathbf{u}}(\mathbf{u})$, are fixed, which facilitates an efficient gradient descent training. For base distribution with trainable parameters, e.g., $q_{\mathbf{u}}(\mathbf{u}; \psi)$, one can reparameterize $\mathbf{u}$ as $\mathbf{u} = T'(\mathbf{u}'; \psi)$, and absorb $T'(\cdot; \psi)$ into the main transformation $T(\cdot; \phi)$, where $\phi$ is the collection of all trainable parameters. Hence, the base distribution becomes $q_{\mathbf{u}'}(\mathbf{u}')$ with only fixed parameters, which do not participate in the gradient descent training. See Section B of the appendix for a practical example.

The applications of normalizing flows can be roughly divided into two categories. If data is available but its distribution is unrevealed (e.g., an image dataset), then one could use a flow-based model to learn the distribution of the data by maximum likelihood estimation, and then make inference on existing data points or generate new data points. Some pioneering works, such as NICE (Dinh et al., 2015), Real NVP (Dinh et al., 2017), and MAF (Papamakarios et al., 2017), have laid the foundation for density estimation using normalizing flows. If a target distribution is given but the sampling method is unknown (e.g., a posterior distribution), then one could approximate the target distribution using variational inference with a flow-based model. Variational inference with normalizing flows was popularized by Rezende & Mohamed (2015). It could be further categorized based on the intended purposes: to make inference on model parameters of interest (Louizos & Welling, 2017), to provide a lower bound for the marginal likelihood of the observed data (Rezende & Mohamed, 2015; Kingma et al., 2016; Tomczak & Welling, 2016; van den Berg et al., 2018), and to construct a proposal distribution for other Monte Carlo sampling methods (Noé et al., 2019; Albergo et al., 2019).

Huang et al. (2018) generalized works of Kingma et al. (2016) and Papamakarios et al. (2017) and proposed neural autoregressive flows, which work as universal approximators for continuous probability distributions. Similarly, Jaini et al. (2019) proposed sum-of-squares polynomial flows and proved the universal approximation property. The universality also applies to spline-based flows (Durkan et al., 2019a;b) as the number of knots used by the spline increases. Chen et al. (2018) introduced continuous normalizing flows by using ordinary differential equations, which were further improved by Grathwohl et al. (2019). Kingma & Dhariwal (2018) proposed Glow, a generative flow model that is able to synthesize and manipulate realistic-looking facial images. Ho et al. (2019) proposed Flow++, which used the variational dequantization technique and achieved a state-of-the-art density estimation performance. Normalizing flows are now considered matured and consists of many works with expanding applicability that we cannot enumerate them all here.

## 3 METHOD

To improve the performance of variational inference, we use normalizing flows to construct a flexible and rich family of distributions, $q_{\mathbf{z}}(\mathbf{z}; \phi)$. That is, we approximate the target distributions, $p(\mathbf{z}|\mathbf{x})$, by minimizing the Kullback-Leibler divergence. As discussed in the introduction, the target distribution is only required up to a multiplicative normalizing constant. In practice, we minimize the negative evidence lower bound,

$$-\text{ELBO}(\phi) = \mathbb{E}_{q_{\mathbf{z}}(\mathbf{z};\phi)}[\log q_{\mathbf{z}}(\mathbf{z}; \phi) - \log p(\mathbf{z}, \mathbf{x})] \tag{1}$$
$$= \mathbb{E}_{q_{\mathbf{u}}(\mathbf{u})}[\log q_{\mathbf{u}}(\mathbf{u}) - \log |\det J_T(\mathbf{u}; \phi)| - \log p(T(\mathbf{u}; \phi), \mathbf{x})],$$

where we absorb all trainable parameters into the transformation $T$; hence $\mathbb{E}_{q_{\mathbf{u}}(\mathbf{u})}[\log q_{\mathbf{u}}(\mathbf{u})]$ is a constant with respect to $\phi$.

In general, the KL divergence or the negative evidence lower bound above does not have a closed-form expression. We compute all expectations analytically whenever possible and approximate the rest by a Monte Carlo estimate otherwise. Then an unbiased gradient estimator can be derived to be used in stochastic gradient-based methods. Let $\{\mathbf{u}_s\}_{s=1}^S$ be a set of samples drawn from the base distribution, $q_{\mathbf{u}}(\mathbf{u})$. Then

$$-\nabla_\phi \text{ELBO}(\phi) \approx -\frac{1}{S} \sum_{s=1}^S [\nabla_\phi \log |\det J_T(\mathbf{u}_s; \phi)| + \nabla_\phi \log p(T(\mathbf{u}_s; \phi), \mathbf{x})].$$

Note that in Equation 1, the expectation is taken with respect to the distribution $q_{\mathbf{z}}(\cdot)$ with parameter $\phi$, which makes the gradient computation problematic. This is solved by using the change of variables formula to free $\phi$ from the expectation. This computational strategy and using Monte Carlo estimation together is referred to as *stochastic gradient variational Bayes* (Kingma & Welling, 2014), also known as *stochastic backpropagation* (Rezende et al., 2014).

For the remainder of this section, we review planar flows in detail and propose a new reparameterization that makes the planar flows free from a pre-existing singularity.

### 3.1 PLANAR FLOWS

We consider the planar flows proposed by Rezende & Mohamed (2015). It is a sequence of transformations of the form

$$f(\mathbf{z}) = \mathbf{z} + \mathbf{v} h(\mathbf{w}^\top \mathbf{z} + b),$$

where $\mathbf{v} \in \mathbb{R}^D$, $\mathbf{w} \in \mathbb{R}^D$, and $b \in \mathbb{R}$ are the parameters, and $h(\cdot)$ is a differentiable activation function such as the hyperbolic tangent. The function $f(\cdot)$ above represents only a single layer of a planar flow model. The layer index $k$ is removed from $\mathbf{v}_k, \mathbf{w}_k, b_k$ and $\mathbf{z}_{k-1}$ for ease of notation.

Planar flows belong to a class of transformations called residual flows, $f(\mathbf{z}) = \mathbf{z} + g(\mathbf{z})$. Geometrically, a displacement vector, $g(\mathbf{z})$, is added to the input vector, $\mathbf{z}$. For planar flows, the displacement vector is in the direction of $\mathbf{v}$ and scaled by $h(\mathbf{w}^\top \mathbf{z} + b)$. Overall, the transformation can be interpreted as a composition of an expansion/contraction in the direction of $\mathbf{w}$ and a shear along the hyperplane $\mathbf{w}^\top \mathbf{z} + b = 0$. Figure 1 demonstrates the geometric effects of the transformation on a circular region centered at the origin with radius 2.

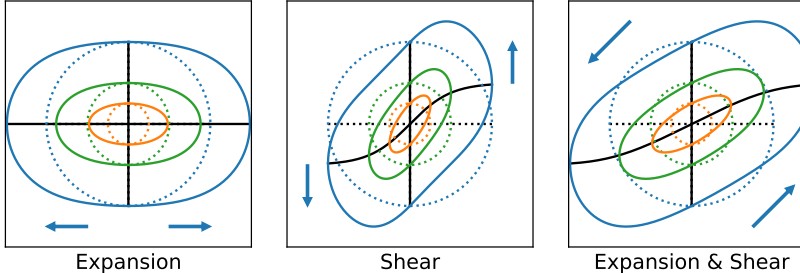

Figure 1: Geometric effects of the planar flow on the $x$-axis, $y$-axis, and three circles of radius 0.5, 1, and 2. From left to right, the parameter $\mathbf{v}$ is $(1,0)^\top$, $(0,1)^\top$, and $(1,1)^\top$ respectively, where $\mathbf{w} = (1,0)^\top$ and $b = 0$ for all cases.

The Jacobian determinant of the transformation is given by:

$$\det J_f(\mathbf{z}) = \det(\mathbf{I} + h'(\mathbf{w}^\top \mathbf{z} + b)\mathbf{v}\mathbf{w}^\top)$$
$$= 1 + h'(\mathbf{w}^\top \mathbf{z} + b)\mathbf{w}^\top \mathbf{v},$$

where the last equality follows from the matrix determinant lemma.

Note that this transformation is not invertible for all values of $\mathbf{w}$ and $\mathbf{v}$. To guarantee invertibility, one could impose an implicit constraint on the parameters:

$$\mathbf{w}^\top \mathbf{v} > -\frac{1}{\sup_x h'(x)}.$$

We follow Rezende & Mohamed (2015) and use $h(x) = \tanh(x)$. Hence, the constraint reduces to

$$\mathbf{w}^\top \mathbf{v} > -1.$$

## 3.2 Reparameterization

To ensure $\mathbf{w}^\top \mathbf{v} > -1$, one can let $\mathbf{w}$ be an unconstrained vector and reparameterize $\mathbf{v}$. Rezende & Mohamed (2015) defined

$$\mathbf{v} = \mathbf{v}' + [m(\mathbf{w}^\top \mathbf{v}') - \mathbf{w}^\top \mathbf{v}']\frac{\mathbf{w}}{||\mathbf{w}||^2}, \tag{2}$$

where $\mathbf{v}'$ is an unconstrained vector and $m(x) = -1 + \log(1 + e^x)$. By doing so, the dot product of $\mathbf{w}$ and $\mathbf{v}$ satisfies the required restriction, i.e., $\mathbf{w}^\top \mathbf{v} = m(\mathbf{w}^\top \mathbf{v}') > -1$, since $m(x) > -1$ for all $x \in \mathbb{R}$. Geometrically, a vector in the direction of $\mathbf{w}$ is added to the unconstrained vector $\mathbf{v}'$ so that the dot product of $\mathbf{w}$ and $\mathbf{v}$ is always greater than $-1$.

While this reparameterization ensures $\mathbf{w}^\top \mathbf{v} > -1$, it is not continuous due to a singularity at $\mathbf{w} = \mathbf{0}$. This singularity is not removable. More specifically, $||\mathbf{v}|| \to \infty$ as $\mathbf{w} \to \mathbf{0}$ for any given $\mathbf{v}'$. Note that not only the size of $\mathbf{v}$ diverges, its direction also changes depending on the trajectory of $\mathbf{w} \to \mathbf{0}$. Figure 2 illustrates how $\mathbf{v}$ diverges as $\mathbf{w}$ spirals inward to $\mathbf{0}$. As the reparameterized $\mathbf{v}$ is sensitive to $\mathbf{w}$ around $\mathbf{0}$, a small update in $\mathbf{w}$ could lead to a huge change in $\mathbf{v}$, and hence an unstable transformation. This non-removable singularity creates a potential for the model to overshoot and get stuck in some undesirable states.

We modify the function $m(\cdot)$ to remove the singularity. The key reason that $\mathbf{v}$ explodes as $\mathbf{w}$ vanishes is because $m(0) \neq 0$. Consider a fixed $\mathbf{v}'$ and let $\mathbf{w} \to \mathbf{0}$. Then the dot product $\mathbf{w}^\top \mathbf{v} = m(\mathbf{w}^\top \mathbf{v}')$ will approach to some nonzero value. Hence, $\mathbf{v}$ must increase its size to maintain the nonzero dot product as $\mathbf{w}$ vanishes. To avoid such explosion, we need $m(0) = 0$. However, simply having $m(\cdot)$ passing the origin is not sufficient, since $\mathbf{v}$ could also explode if $m(\cdot)$ decays to zero too slowly. Rewrite Equation 2 as

$$\mathbf{v} = \mathbf{v}' + \left[\frac{1}{||\mathbf{w}||}m(||\mathbf{w}||\hat{\mathbf{w}}^\top \mathbf{v}') - \hat{\mathbf{w}}^\top \mathbf{v}'\right]\hat{\mathbf{w}}, \tag{3}$$

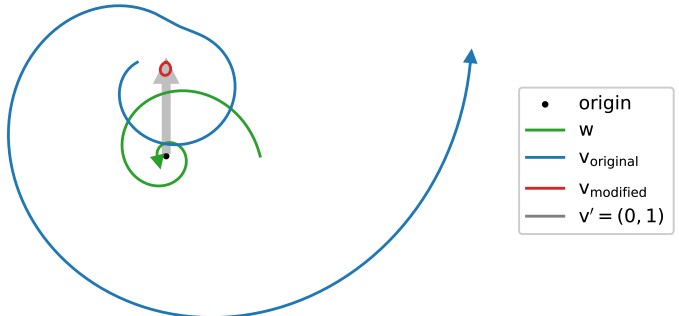

Figure 2: The original and modified reparametrized $\mathbf{v}$ as $\mathbf{w} \to \mathbf{0}$ along an equiangular spiral with polar coordinates given by $r = e^{-\varphi/4}$, where $\varphi \in [0, 3\pi]$ and the unconstrained $\mathbf{v}'$ is fixed at $(0, 1)^\top$. The original reparametrized $\mathbf{v}$ diverges as $\varphi$ increases, whereas the modified reparametrized $\mathbf{v}$ equals to $\mathbf{v}'$ when $\varphi \in [0, \pi] \cup [2\pi, 3\pi]$ and differs slightly from $\mathbf{v}'$ when $\varphi \in (\pi, 2\pi)$.

where $\hat{\mathbf{w}}$ is the unit vector of $\mathbf{w}$. We see that $m(x) \in \mathcal{O}(x)$ as $x \to 0$ is sufficient to remove the singularity at $\mathbf{w} = \mathbf{0}$. Note that the role of $m(\cdot)$ is to map $\mathbf{w}^\top \mathbf{v}' \in (-\infty, \infty)$ to $\mathbf{w}^\top \mathbf{v} \in (-1, \infty)$, which in turn reparameterizes the unconstrained $\mathbf{v}'$ to the feasible values. To achieve a minimal reparameterization for $\mathbf{v}'$, we consider

$$m(x) = \begin{cases} x & \text{if } x \geq 0 \\ e^x - 1 & \text{if } x < 0. \end{cases}$$

Despite being a piecewise function, the modified $m(\cdot)$ is continuously differentiable in the whole real line. With this modification, $\mathbf{w} = \mathbf{0}$ in Equation 3 becomes a removable singularity (see Figure 2 for an illustration). In practice, to eliminate the singularity completely, rather than using Equation 3, we simplify the expression and define $\mathbf{v}$ by cases:

$$\mathbf{v} = \begin{cases} \mathbf{v}' & \text{if } \mathbf{w}^\top \mathbf{v}' \geq 0 \\ \mathbf{v}' + [\exp(\mathbf{w}^\top \mathbf{v}') - 1 - \mathbf{w}^\top \mathbf{v}'] \frac{\mathbf{w}}{||\mathbf{w}||^2} & \text{if } \mathbf{w}^\top \mathbf{v}' < 0. \end{cases}$$

Note that, (i) no reparameterization is required if $\mathbf{w}^\top \mathbf{v}' \geq 0$; (ii) the dot product $\mathbf{w}^\top \mathbf{v}'$ is sign-preserving under this reparameterization; (iii) the reparameterization for $\mathbf{v}$ is now continuously differentiable. These properties make the geometric effects of the transformation more transparent with respect to the underlying unconstrained parameters. For example, $\mathbf{w}^\top \mathbf{v}' \geq 0$ indicates an expansion and the displacement vector is in the direction of $\mathbf{v}'$.

The parameter initialization also benefits from this simple reparameterization. A typical practice is to randomly initialize all trainable parameters $(\mathbf{w}, \mathbf{v}'$ and $b)$ to some values around $\mathbf{0}$. For the new reparameterization, we have either $\mathbf{v} = \mathbf{v}'$ or $\mathbf{v} \approx \mathbf{v}'$, whereas in the original reparameterization, $\mathbf{v}$ is sensitive to $\mathbf{w}$ around $\mathbf{0}$, which leads to an unstable initial flow network.

## 4 EXPERIMENTS

We conduct three experiments to demonstrate the effectiveness of the singularity-free planar flows. More specifically, we compare the old and the new reparameterizations for planar flows in terms of model training performance. For simplicity, we refer to the planar flow with the new (old) reparameterization as the new (old) planar flow.

To ensure a fair comparison, we use the same initial parameters for the new and old planar flows such that they have the same initial output distributions. See Section A of the appendix for more details about parameter initialization.

### 4.1 TOY DISTRIBUTIONS

Rezende & Mohamed (2015) demonstrated the representative power of planar flows using some 2D toy distributions. We use the same set of toy distributions but with an extra decay term for the

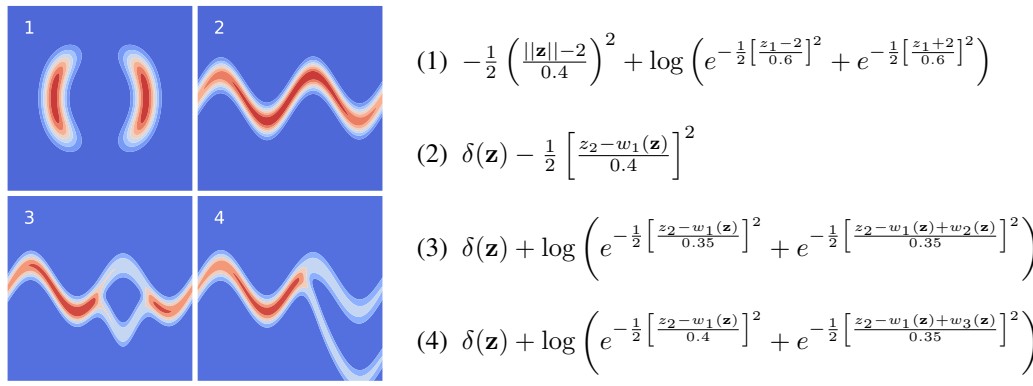

$$(1) \quad -\frac{1}{2}\left(\frac{||\mathbf{z}||-2}{0.4}\right)^2 + \log\left(e^{-\frac{1}{2}\left[\frac{z_1-2}{0.6}\right]^2} + e^{-\frac{1}{2}\left[\frac{z_1+2}{0.6}\right]^2}\right)$$

$$(2) \quad \delta(\mathbf{z}) - \frac{1}{2}\left[\frac{z_2 - w_1(\mathbf{z})}{0.4}\right]^2$$

$$(3) \quad \delta(\mathbf{z}) + \log\left(e^{-\frac{1}{2}\left[\frac{z_2-w_1(\mathbf{z})}{0.35}\right]^2} + e^{-\frac{1}{2}\left[\frac{z_2-w_1(\mathbf{z})+w_2(\mathbf{z})}{0.35}\right]^2}\right)$$

$$(4) \quad \delta(\mathbf{z}) + \log\left(e^{-\frac{1}{2}\left[\frac{z_2-w_1(\mathbf{z})}{0.4}\right]^2} + e^{-\frac{1}{2}\left[\frac{z_2-w_1(\mathbf{z})+w_3(\mathbf{z})}{0.35}\right]^2}\right)$$

Figure 3: $\log \tilde{p}(\mathbf{z})$ of the toy distributions, in which $w_1(\mathbf{z}) = \sin(\frac{\pi z_1}{2})$, $w_2(\mathbf{z}) = 3e^{-\frac{1}{2}\left[\frac{z_1-1}{0.6}\right]^2}$, $w_3(\mathbf{z}) = 3\sigma(\frac{z_1-1}{0.3})$, and $\sigma(x) = 1/(1 + e^{-x})$. A decay term, $\delta(\mathbf{z}) = -\frac{1}{2}\left[\frac{z_1}{5}\right]^2$, is added to the last three cases so that they become proper probability distributions.

last three cases so that they become proper probability distributions. See Figure 3 for the modified distributions and unnormalized log-densities, $\log \tilde{p}(\mathbf{z})$.

We use the new and old planar flows to approximate each toy distribution. We consider flows with 2, 4, 8, 16, and 32 layers. A general Gaussian distribution, $\mathcal{N}(\boldsymbol{\mu}, \boldsymbol{\Sigma})$, is used as the base distribution. To facilitate an efficient gradient descent training, we absorb the parameters of the base distribution into the flow transformation by prepending an extra invertible linear layer to the main flow. Section B of the appendix gives more details of the prepended linear layer.

We minimize the KL divergence,

$$\mathbb{E}_{q_{\mathbf{u}}(\mathbf{u})}[\log q_{\mathbf{u}}(\mathbf{u}) - \log|\det J_T(\mathbf{u}; \boldsymbol{\phi})| - \log \tilde{p}(T(\mathbf{u}; \boldsymbol{\phi}))] + \log \mathrm{const},$$

where the normalizing constant is computed by integrating $\tilde{p}(\mathbf{z})$. Since we could generate random points freely from the base distribution, it can be considered as a training dataset with unlimited data points. We train the flow model using stochastic gradient descent with batch size 250 for $500k$ parameter updates ($k$, short for *thousand* for simplicity). For the optimization algorithm, we use Adam (Kingma & Ba, 2015) with initial learning rate $0.001$. Then we decay the learning rate by a multiplicative factor of $0.95$ every $10k$ parameter updates.

Starting from $50k$, we evaluate the KL divergence with one million Gaussian points every $10k$ parameter updates. Figure 4 shows the KL divergences aggregated from 100 replicates. We see that the new planar flows converge faster and to a better place in most settings. The gap between the two flows decreases as the number of layers increases. This is because there is no much room for improvement when the KL divergences of both flows converge to zero. In general, the new planar flows are more efficient. To achieve the same level of approximation quality, the old planar flows need more layers than the new planar flows.

## 4.2 BAYESIAN REGRESSIONS

We consider linear regression and logistic regression in Bayesian paradigm and use planar flows to approximate the posterior distributions of the regression parameters. A horseshoe-like prior is used for the regression parameters,

$$p(\beta_j) = \frac{1}{2\pi s} \log\left[1 + \frac{1}{(\beta_j/s)^2}\right],$$

where $s > 0$ is a scale parameter. We call this class of priors spike priors. Similar to horseshoe priors, the spike priors have heavy Cauchy-like tails and a pole at zero, i.e., tails decaying like $\beta^{-2}$ and $\lim_{\beta \to 0} p(\beta) = \infty$ (Carvalho et al., 2010). Unlike the hierarchical construction of the horseshoe priors, the spike priors have a closed-form density, which is more convenient for variational inference. In this experiment, we fix the scale parameter $s = 0.1$.

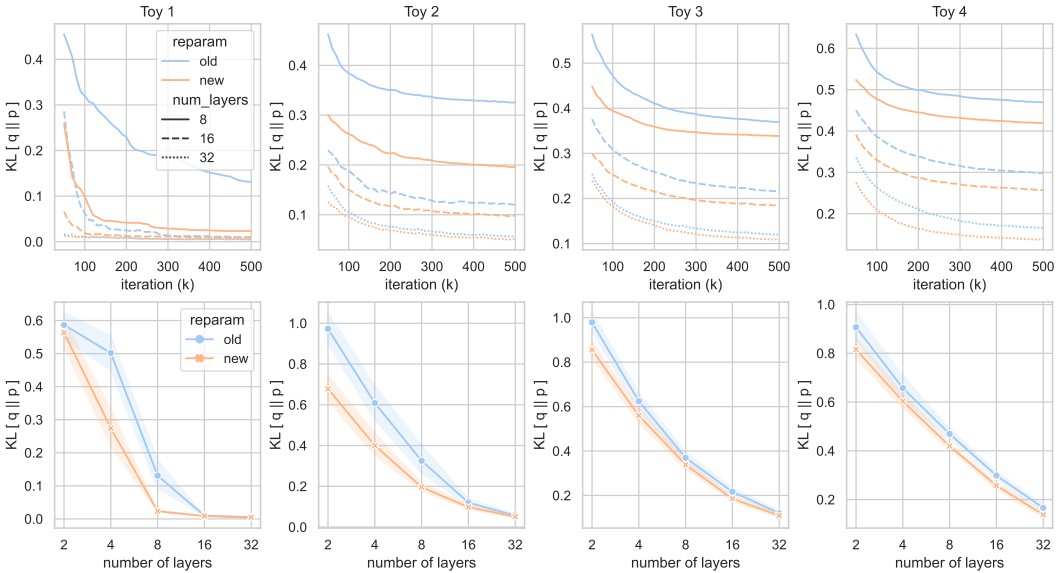

Figure 4: The first row shows the KL divergence evaluated every $10k$ parameter updates. The second row gives the KL divergence of the final trained models. Results are aggregated from 100 replicates.

Let

$$\eta_i = \beta_1 x_{i1} + \beta_2 x_{i2} + \sum_{j=3}^{10} \beta_j x_{ij}.$$

For linear regression, we consider $y_i = \eta_i + \varepsilon_i$, where $\varepsilon_i \sim \mathcal{N}(0, 1)$ is assumed known. For logistic regression, we consider $y_i \sim \mathrm{Ber}(p_i)$ and $\log(\frac{p_i}{1-p_i}) = \eta_i$.

For each replicate, the true values of $(\beta_1, \beta_2)$ are sampled from $\mathrm{Unif}(-1, 1)$. We set $\beta_j = 0$ for $j \geq 3$ to create sparsity. The covariate vectors, $(x_{i1}, \ldots, x_{i,10})$, are generated from a multivariate Gaussian distribution with zero mean, unit marginal variance, and AR-1 correlation structure with parameter $0.5$. The data size for linear regression and logistic regression are set to 10 and 20 respectively. Note that the data $(y_i, \mathbf{x}_i)$ and the true values of the regression parameters $\boldsymbol{\beta}$ are randomly generated for each replicate. Hence, the posterior distributions of the regression parameters are different across replicates.

We ignore the normalizing constant and minimize the negative evidence lower bound. As in the previous experiment, we use a general Gaussian distribution as the base distribution; train the flow model using stochastic gradient descent with batch size 250 for $500k$ parameter updates; use the Adam algorithm with initial learning rate 0.001 and decay the learning rate by a multiplicative factor of 0.95 every $10k$ parameter updates.

We evaluate $-$ELBO with one million Gaussian points every $10k$ parameter updates. Figure 5 shows the estimates of $-$ELBO and the difference between the two planar flows. The results are aggregated from 100 replicates. Again, we see that the new planar flows converge faster and to a better place.

### 4.3 Variational Autoencoder

We train variational autoencoders with normalizing flows using the binarized MNIST digit dataset (LeCun et al., 1998). The dataset contains $60,000$ training and $10,000$ test images of ten handwritten digits. We consider the old and new planar flows, the inverse autoregressive flows (IAFs, Kingma et al., 2016), the Sylvester normalizing flows (SNFs, van den Berg et al., 2018), and the neural spline flows (NSFs, Durkan et al., 2019b). For all flows, we use latent dimension $D = 20$ and the same variational autoencoder architecture, which is illustrated with a figure in Section C of the appendix.

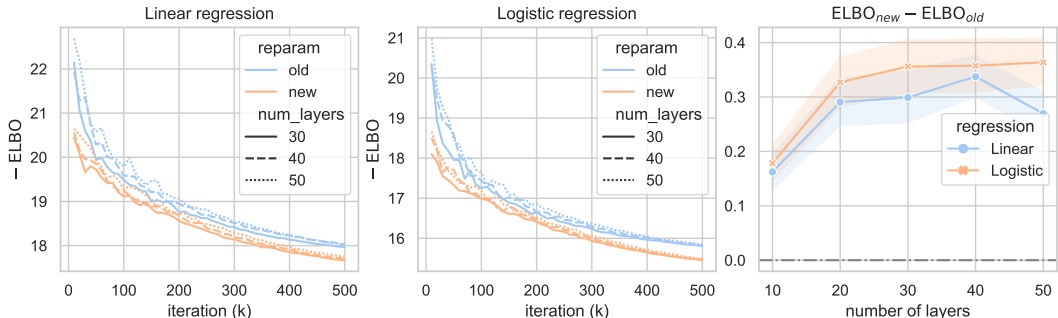

Figure 5: The negative evidence lower bound evaluated every $10k$ parameter updates. The third figure shows the difference between the two planar flows when the training is complete.

The framework of variational autoencoders was introduced by Kingma & Welling (2014) and Rezende et al. (2014). It consists of two parts, an inference network and a generative network. The inference network takes in a data point and compresses it into a low-dimensional latent space, whereas the generative network samples a random point from the encoded latent space and converts it back to the data space.

As a probabilistic graphical model, the generative network models a joint distribution $p_{\boldsymbol{\theta}}(\mathbf{z}, \mathbf{x}) = p_{\boldsymbol{\theta}}(\mathbf{z}|\mathbf{x})p_{\boldsymbol{\theta}}(\mathbf{z})$, where $p_{\boldsymbol{\theta}}(\mathbf{z})$ is the prior distribution over the latent space and $p_{\boldsymbol{\theta}}(\mathbf{x}|\mathbf{z})$ is the likelihood. The primary goal is to maximize the marginal likelihood, $p_{\boldsymbol{\theta}}(\mathbf{x}) = \int p_{\boldsymbol{\theta}}(\mathbf{z}|\mathbf{x})p_{\boldsymbol{\theta}}(\mathbf{z})d\mathbf{z}$. However, the integral becomes intractable even when the likelihood is modeled by a moderately complicated neural network. To solve this issue, Kingma & Welling (2014) introduced the inference network, $q_{\boldsymbol{\phi}}(\mathbf{z}|\mathbf{x})$, to approximate the intractable true posterior, $p_{\boldsymbol{\theta}}(\mathbf{z}|\mathbf{x})$. Then the evidence lower bound, a lower bound for the marginal likelihood, could be used as the optimization criterion,

$$\log p_{\boldsymbol{\theta}}(\mathbf{x}) = \underbrace{D_{\mathrm{KL}}[q_{\boldsymbol{\phi}}(\mathbf{z}|\mathbf{x})\|p_{\boldsymbol{\theta}}(\mathbf{z}|\mathbf{x})]}_{\geq 0} + \underbrace{\mathbb{E}_{q_{\boldsymbol{\phi}}(\mathbf{z}|\mathbf{x})}[\log p_{\boldsymbol{\theta}}(\mathbf{z}, \mathbf{x}) - \log q_{\boldsymbol{\phi}}(\mathbf{z}|\mathbf{x})]}_{\mathrm{ELBO}(\boldsymbol{\theta},\boldsymbol{\phi};\mathbf{x})}.$$

By maximizing $\mathrm{ELBO}(\boldsymbol{\theta}, \boldsymbol{\phi}; \mathbf{x})$ over $\boldsymbol{\theta}$ and $\boldsymbol{\phi}$ at the same time, the marginal likelihood is improved and the intractable posterior is learned.

Normalizing flows can be used in the inference networks of variational autoencoders to better approximate the posterior distributions of the latent variables (Rezende & Mohamed, 2015). A better posterior approximation provides a tighter bound for the marginal likelihood, which leads to a more accurate maximum likelihood estimate. More specifically, the diagonal Gaussian variational distribution, $q_{\boldsymbol{\phi}}(\mathbf{z}|\mathbf{x}) = \mathcal{N}(\mathbf{z}|\boldsymbol{\mu}_{\boldsymbol{\phi}}(\mathbf{x}), \mathrm{diag}(\boldsymbol{\sigma}_{\boldsymbol{\phi}}^2(\mathbf{x})))$, can be extended to a more flexible distribution constructed by a flow model,

$$q_{\boldsymbol{\phi}}(\mathbf{z}|\mathbf{x}) = q_{\mathbf{u}}(\mathbf{u}; \boldsymbol{\phi})|\det J_T(\mathbf{u}; \boldsymbol{\phi})|^{-1},$$

where

$$\mathbf{z} = T(\mathbf{u}) \quad \text{and} \quad q_{\mathbf{u}}(\mathbf{u}; \boldsymbol{\phi}) = \mathcal{N}(\mathbf{u}|\boldsymbol{\mu}_{\boldsymbol{\phi}}(\mathbf{x}), \mathrm{diag}(\boldsymbol{\sigma}_{\boldsymbol{\phi}}^2(\mathbf{x}))).$$

The transformation of an IAF layer is given by $f(\mathbf{z}) = \boldsymbol{\sigma} \odot \mathbf{z} + (\mathbf{1} - \boldsymbol{\sigma}) \odot \mathbf{m}$, where $\boldsymbol{\sigma} = \mathrm{sigmoid}(\mathbf{s})$ and $[\mathbf{m}, \mathbf{s}] = \mathrm{AutoregressiveNN}(\mathbf{z})$. For the NSF layer, we use the same autoregressive structure as in the IAF layer and the affine transformation $f(\cdot)$ is replaced with elementwise spline functions with $K = 5$ bins and tail bound $B = 3$. We follow Kingma et al. (2016) and use a two-layer MADE (Germain et al., 2015) to implement the autoregressive network. However, for the activation function of MADE, we choose ReLU over ELU (Clevert et al., 2016), since ReLU leads to better results in our experiment. We use 40 units for both hidden layers of MADE. Under this setting, the parameter sizes of IAFs and NSFs are approximately 50 and 170 times greater than that of planar flows. Section D provides more details about the number of parameters.

The transformation of an SNF layer is given by $f(\mathbf{z}) = \mathbf{z} + \mathbf{Q}\mathbf{R}\tanh(\tilde{\mathbf{R}}\mathbf{Q}^{\top}\mathbf{z} + \mathbf{b})$, where $\mathbf{R}$, $\tilde{\mathbf{R}} \in \mathbb{R}^{M \times M}$ are lower triangular, $\mathbf{Q} \in \mathbb{R}^{D \times M}$ contains $M$ orthonormal column vectors, $\mathbf{b} \in \mathbb{R}^D$, and $M \leq D$. The orthogonality is obtained by using an iterative procedure proposed by Björck &

Bowie (1971): $\mathbf{Q}^{(k+1)} = \mathbf{Q}^{(k)} \left( \mathbf{I} + \frac{1}{2}(\mathbf{I} - \mathbf{Q}^{(k)\top}\mathbf{Q}^{(k)}) \right)$. This step makes the SNFs much more computationally demanding than the planar flows. SNFs can be considered as a generalization of planar flows but not in a strict sense, since they do not reduce to planar flows when $M = 1$. We set $M = 2$ to double the bottleneck size.

The training details are given in Section E of the appendix. For the final trained models, we estimate the marginal likelihood with all $10,000$ test images using an importance sampling technique proposed by Rezende et al. (2014). The marginal log-likelihood of a data point is given by

$$\log p_{\boldsymbol{\theta}}(\mathbf{x}) \approx \log \frac{1}{S} \sum_{s=1}^{S} \frac{p_{\boldsymbol{\theta}}(\mathbf{x}, \mathbf{z}_s)}{q_{\boldsymbol{\phi}}(\mathbf{z}_s|\mathbf{x})},$$

where the Monte Carlo estimate size $S$ is set to 500 for better accuracy. The ELBO$(\boldsymbol{\theta}, \boldsymbol{\phi}; \mathbf{x})$ is re-estimated with $S = 500$ for each test image. Then the variational gap is computed by

$$D_{\mathrm{KL}}[q_{\boldsymbol{\phi}}(\mathbf{z}|\mathbf{x})\|p_{\boldsymbol{\theta}}(\mathbf{z}|\mathbf{x})] = \log p_{\boldsymbol{\theta}}(\mathbf{x}) - \mathrm{ELBO}(\boldsymbol{\theta}, \boldsymbol{\phi}; \mathbf{x}).$$

Figure 6 summarizes the performance of each model as the number of flow layers increases. The results are aggregated from 10 replicates. As expected, the new planar flows approximate the posteriors better than the old planar flows, which leads to better evidence lower bounds and marginal likelihoods. Both the new planar flows and the SNFs outperform the IAFs and NSFs in posterior approximation. While the IAFs and NSFs employed in this study have much larger numbers of parameters compared to the planar flows, it is essential to note that their sizes remain relatively small in the context of the neural network architecture. Also, training of such intricate models with deep layers may require extra attention. Consequently, the results do not reflect their optimal performance and are solely used for comparison in terms of parameter efficiency.

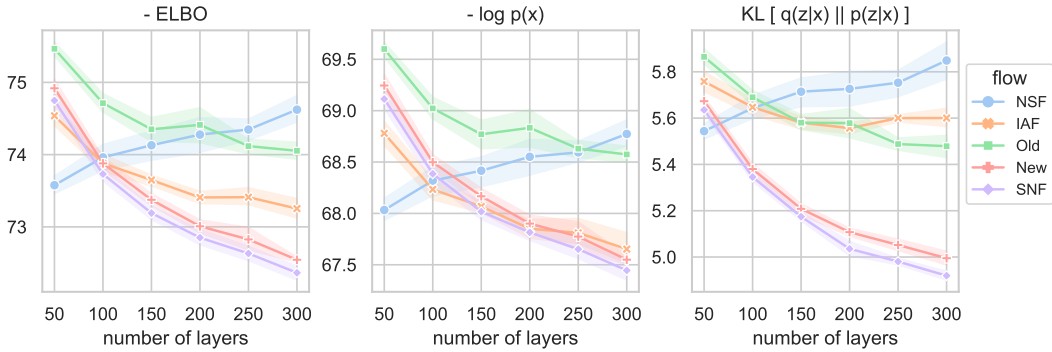

Figure 6: The performance of each model as the number of flow layers increases.

## 5 CONCLUSION

We proposed a new reparameterization to make the planar flows free from the pre-existing singularity. The presence of this singularity renders certain distributions within the variational family practically unattainable, thereby limiting the expressiveness of planar flows. By eliminating the singularity and employing a continuously differentiable reparameterization, distributions within the whole variational family can be explored smoothly during training, which leads to better approximation in general.

Our experiments show that the new reparameterization improves the planar flows significantly in various variational inference tasks. The performance of the singularity-free planar flows is comparable to the IAFs, NSFs, and SNFs, which are much larger in parameter size and more computationally demanding.

However, the expressiveness of planar flows is still limited due to the simple transformation per layer. Many layers need to be stacked to output flexible distributions. In choosing simplicity, our intention was to provide a clear and accessible solution to general variational problems, particularly in scenarios where ease of implementation and computational efficiency are crucial.

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

## A    PARAMETER INITIALIZATION

We consider the default initialization method used for the linear layer in Pytorch 2.0; that is, we initialize all elements of the planar flow parameters $\{\mathbf{w}_k, \mathbf{v}_k, b_k\}_{k=1}^{K}$ independently from a uniform distribution,

$$\text{Unif}\left(-\frac{1}{\sqrt{D}}, \frac{1}{\sqrt{D}}\right),$$

where $D$ is the dimension of the input random vector $\mathbf{u}$. Then for the new and old planar flows, we compute the corresponding unconstrained parameters $\{\mathbf{v}_k'\}_{k=1}^{K}$ with the same values of $\{\mathbf{w}_k, \mathbf{v}_k\}_{k=1}^{K}$. Hence, the new and old planar flows share the same initial output distributions.

The parameters of the prepended linear layers in the first two experiments and the masked linear layers of MADEs in the third experiment are initialized the same way from a uniform distribution.

## B    PREPENDED LINEAR LAYER

Consider a normalizing flow, $\mathbf{z} = T(\mathbf{u})$, where the base distribution $\mathbf{u} \sim \mathcal{N}(\boldsymbol{\mu}, \boldsymbol{\Sigma})$. To facilitate an efficient gradient descent training, one could reparameterize $\mathbf{u}$ as

$$\mathbf{u} = T_0(\mathbf{u}') = \boldsymbol{\mu} + \boldsymbol{L}\mathbf{u}',$$

where $\mathbf{u}' \sim \mathcal{N}(\mathbf{0}, \mathbf{I})$ and $\boldsymbol{L}$ is given by the Cholesky decomposition $\boldsymbol{\Sigma} = \boldsymbol{L}\boldsymbol{L}^{\top}$.

Let $T \leftarrow T \circ T_0$ and $\mathbf{u} \leftarrow \mathbf{u}'$. Then the parameters of the base distribution, $\{\boldsymbol{\mu}, \boldsymbol{\Sigma}\}$, are absorbed into the flow transformation as $\{\boldsymbol{\mu}, \boldsymbol{L}\}$. Now the base distribution contains only fixed parameters, which do not participate in the gradient descent training.

In terms of the architecture of the flow model, it is equivalent to using $\mathcal{N}(\mathbf{0}, \mathbf{I})$ as the base distribution, and then prepending an extra invertible linear layer to the main flow. The weight matrix of the prepended linear layer is lower triangular with positive diagonal entries. We reparameterize the diagonal entries with the following function to ensure positivity,

$$f(x) = \begin{cases} x + 1 & \text{if } x \geq 0 \\ e^x & \text{if } x < 0. \end{cases}$$

We use a linear function for the positive domain instead of a pure exponential function to reduce the chance of overflow. Note that $f(0) = 1$. Hence, positive and negative raw diagonal entries imply expansion and contraction respectively, which occur with equal chances under our symmetrical uniform parameter initialization.

## C    ARCHITECTURE OF VAE

Figure 7 illustrates the architecture of the variational autoencoders used in the experiment.

## D    NUMBER OF PARAMETERS

We compute the number of parameters for each flow used in the variational autoencoders. Let $D$ be the dimension of the latent space.

For planar flows, the number of parameters per flow layer is given by $2D + 1$. For Sylvester flows, the number of parameters per flow layer is given by $MD + M(M + 1) + M$, where $M = 2$ is the number of orthonormal column vectors in matrix $\mathbf{Q}$.

For IAFs and NSFs, the number of parameters is much larger, since a two-layer of MADE is used to implement the autoregressive network, in which $H = 40$ is used for the number of units in both hidden layers of MADE. We assume that the autoregressive masking removes approximately half of the weights. Hence, the number of *weight* parameters per flow layer is given by $\frac{1}{2}(HD + H^2 + 2HD)$ for IAFs; and $\frac{1}{2}(HD + H^2 + (3K - 1)HD)$ for NSFs, where $K = 5$ is the number of bins used for the spline functions.

Table 1 summarizes the number of parameters per flow layer for each flow used in the variational autoencoders.

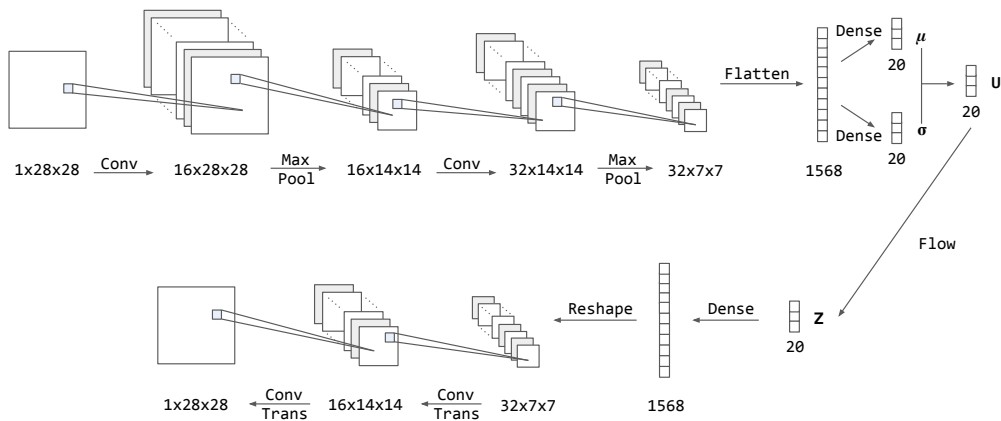

Figure 7: The architecture of the variational autoencoders.

Table 1: Number of parameters per flow layer.

| Flow | #{weight} | #{bias} | $D = 20$ |
|---|---|---|---|
| Planar | $2D + 1$ | $1$ | 41 |
| Sylvester | $MD + M(M + 1)$ | $M$ | 48 |
| IAF | $\frac{1}{2}(HD + H^2 + 2HD)$ | $2H + 2D$ | 2120 |
| NSF | $\frac{1}{2}(HD + H^2 + (3K - 1)HD)$ | $2H + (3K - 1)D$ | 7160 |

## E  TRAINING DETAILS FOR THE VAE EXPERIMENT

The loss function per data point, $-\text{ELBO}(\boldsymbol{\theta}, \boldsymbol{\phi}; \mathbf{x})$, can be decomposed into two interpretable terms as follows,

$$
\begin{aligned}
-\text{ELBO}(\boldsymbol{\theta}, \boldsymbol{\phi}; \mathbf{x}) &= \mathbb{E}_{q_{\boldsymbol{\phi}}(\mathbf{z}|\mathbf{x})}[\log q_{\boldsymbol{\phi}}(\mathbf{z}|\mathbf{x}) - \log p_{\boldsymbol{\theta}}(\mathbf{z}, \mathbf{x})] \\
&= \mathbb{E}_{q_{\boldsymbol{\phi}}(\mathbf{z}|\mathbf{x})}[\log q_{\boldsymbol{\phi}}(\mathbf{z}|\mathbf{x}) - \log p_{\boldsymbol{\theta}}(\mathbf{z}) - \log p_{\boldsymbol{\theta}}(\mathbf{x}|\mathbf{z})] \\
&= \underbrace{D_{\text{KL}}[q_{\boldsymbol{\phi}}(\mathbf{z}|\mathbf{x})\|p_{\boldsymbol{\theta}}(\mathbf{z})]}_{\text{Regularization term}} + \underbrace{\mathbb{E}_{q_{\boldsymbol{\phi}}(\mathbf{z}|\mathbf{x})}[-\log p_{\boldsymbol{\theta}}(\mathbf{x}|\mathbf{z})]}_{\text{Reconstruction error}}.
\end{aligned}
$$

To prevent the training from getting stuck in some undesirable stable equilibrium, we use annealing as suggested by Bowman et al. (2016) and Sønderby et al. (2016). Specifically, we add a weight to the regularization term at training time,

$$
-\text{ELBO}(\boldsymbol{\theta}, \boldsymbol{\phi}; \mathbf{x}) = \beta_t D_{\text{KL}}[q_{\boldsymbol{\phi}}(\mathbf{z}|\mathbf{x})\|p_{\boldsymbol{\theta}}(\mathbf{z})] + \mathbb{E}_{q_{\boldsymbol{\phi}}(\mathbf{z}|\mathbf{x})}[-\log p_{\boldsymbol{\theta}}(\mathbf{x}|\mathbf{z})],
$$

where $\beta_t$ increases linearly from 0 to 1 over the first 100 epochs.

We use stochastic gradient descent with batch size 250 for 500 epochs (240 parameter updates per epoch). The Monte Carlo estimate size $S$ for the expectation is set to 1, since the batch size is large enough Kingma & Welling (2014). We use the Adam algorithm with initial learning rate 0.001. The learning rate is controlled dynamically: if $-\text{ELBO}(\boldsymbol{\theta}, \boldsymbol{\phi}; \mathbf{x})$ on the $10,000$ test images does not decrease for 10 epochs, then reduce the learning rate by a multiplicative factor of 0.75. After the 500 epochs training, the one with the lowest test loss is kept as the final model.

## F  IMAGES GENERATED FROM THE TRAINED VAES

Images generated from the trained VAEs are shown in Figure 8.

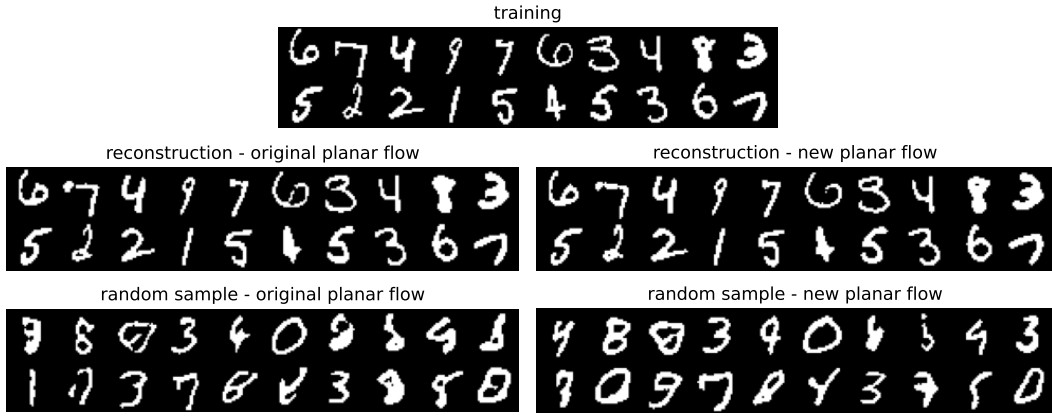

Figure 8: Images generated from the trained VAEs.

## G ADDITIONAL VAE EXPERIMENT WITH HIGHER LATENT DIMENSION

We also run the VAE experiment with a higher latent dimension $D = 40$. Similar results are observed and shown in Figure 9.

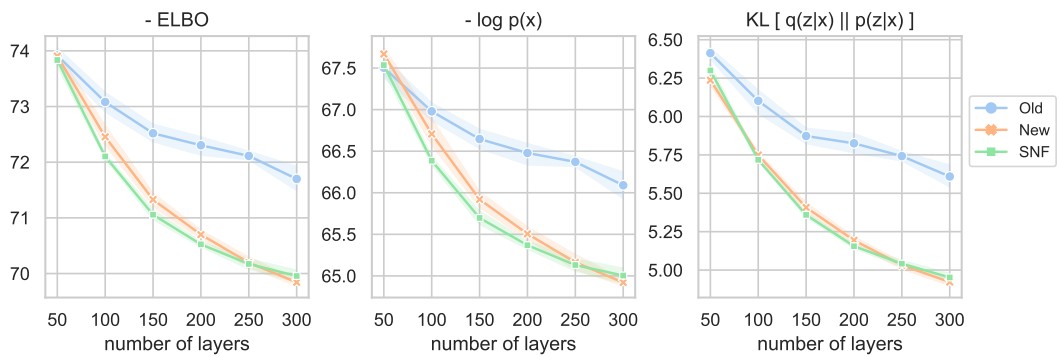

Figure 9: The performance as the number of flow layers increases.

## H ENVIRONMENT INFORMATION FOR EXPERIMENTS

All experiments were performed on Red Hat Enterprise Linux 9.2 (Plow) (x86_64) with Intel(R) Xeon(R) Gold and NVIDIA A100 80GB PCIe. The Python version is 3.11.5. The PyTorch version is 2.0.1 and the CUDA used to build PyTorch is 11.7. For the first two experiments, we trained the flow models with CPUs only. For the third experiment, we used GPUs to accelerate the training of variational autoencoders.

