# OpenReview forum: "Variational Inference with Singularity-Free Planar Flows"
_ICLR.cc/2024/Conference — Submitted to ICLR 2024_

### Official Review · Reviewer_ndCL · 2023-10-26

**Soundness:** 2 fair
**Presentation:** 3 good
**Contribution:** 3 good
**Rating:** 5
**Confidence:** 4

**Summary:**

The work introduced a new reparameterization scheme for the training parameters of planar flows, effectively eliminating the initial singularity issues. This new approach ensures that the parameter $v$ does not escalate to infinity as $||w||$ approaches zero, potentially enhancing the stability during the training of Planar flow.  Empirical results demonstrate that the new parameterization scheme improves the performance of planar flows in variational inference applications.

**Strengths:**

This work delivers a concise and straightforward description of a well-defined problem, and the logic behind the proposed solution is easy to follow.

The solution itself, while simple, effectively addresses a known issue in planar flow, representing a practical contribution that the community is likely to find useful.

**Weaknesses:**

- The primary concern with this work lies in the significance of the problem it addresses.  While the problem focused in the paper is very well-defined (as mentioned in the Strength section), the importance of this problem is hard to justify for the following 2 reasons:
1. Planar flow, the focus of this work, is one of the earliest methods introduced in the normalizing flow literature, and numerous advanced normalizing flows such as affine coupling flows, neural spline flows, etc., have been developed, showcasing superior performance.  Given this context, the contribution of this work appears quite limited.  To justify its contribution,  it would be imperative to demonstrate that the modifications introduced significantly enhance the performance of planar flow. Ideally, it should now be on par with, or outperform, the more recent flow methods. This point leads to the second concern.

2. The empirical experiments conducted in the study lack a comprehensive comparison with a variety of recent flow methods. Moreover, the paper only delves into variational inference applications, omitting performance evaluations on density estimation and the quality of synthetic data generation. To solidify the work's standing and importance, an expansion of the experimental section to include these aspects is necessary.

- Furthermore, this work falls short in providing a thorough and rigorous exploration of the singularity issue. The connection between the presence of a singularity and its detrimental impact on the performance of planar flow remains unclear to readers. The paper asserts that the proposed approach enhances the stability of model training, setting the expectation for a detailed analysis.

To meet this expectation, the paper should ideally include a comparative analysis of the gradient variance in planar flow, employing both the original and the new parameterization schemes. Additionally, a theoretical examination of the training dynamics in a deep planar flow setup would contribute to a more comprehensive understanding. The paper should also offer a conclusive answer to whether issues such as gradient explosion or vanishing are effectively circumvented with the proposed modifications. Without these elements, the paper's exploration of the singularity issue feels incomplete, leaving readers with unanswered questions and a lack of clarity on the significance of the proposed solution.

**Questions:**

no questions.

---

> ### Author Response · Authors · 2023-11-21
>
> # Response to Reviewer ndCL
>
> Thank you for your review and recognizing the practicality of our solution. For your concern on the significance of the problem it addresses, we reply to all your comments below.
>
> > **Weakness 1**: Planar flow, the focus of this work, is one of the earliest methods introduced in the normalizing flow literature, and numerous advanced normalizing flows such as affine coupling flows, neural spline flows, etc., have been developed, showcasing superior performance. Given this context, the contribution of this work appears quite limited. To justify its contribution, it would be imperative to demonstrate that the modifications introduced significantly enhance the performance of planar flow. Ideally, it should now be on par with, or outperform, the more recent flow methods. This point leads to the second concern.
>
> **A1**: We agree that the more recent flow methods are powerful in specific tasks. However, the expressiveness and scalability of the these flow methods come with a price — they are more intricately designed and require extra attention on parameter tuning/training, which can be inefficient in solving some variational inference tasks. This inefficiency is more severe when they are run on CPU-only devices.
>
> While we agree that intricate models with fine-tuned parameters are powerful in specific tasks, we believe that simplicity can be a strength. Our objective was to provide a clear and accessible solution to general variational problems, where ease of implementation and computational efficiency are crucial, e.g., sampling from a customize distribution, posterior inference in Bayesian statistical regression, and low-dimensional VAE.
>
>
> > **Weakness 2**: The empirical experiments conducted in the study lack a comprehensive comparison with a variety of recent flow methods. Moreover, the paper only delves into variational inference applications, omitting performance evaluations on density estimation and the quality of synthetic data generation. To solidify the work's standing and importance, an expansion of the experimental section to include these aspects is necessary.
>
> **A2**: Thank you for your suggestion. We have conducted an extra comparison with a recent flow methods, neural spline flows (NSFs). The updated comparison reaffirms the parameter efficiency of the new planar flows. We have included the results of the NSFs experiment in Figure 6, and a table summarizing the parameter counts is now available in Appendix D in the updated version.
>
> As indicated by our title, *Variational Inference with Singularity-Free Planar Flows*, our focus is solely on variational inference. One reason for this focus is that the inversion of the planar flow does not have a closed form, making it less attractive for density estimation and data generation.

---

> > ### Author Response · Authors · 2023-11-21
> >
> > > **Comment 1**: Furthermore, this work falls short in providing a thorough and rigorous exploration of the singularity issue. The connection between the presence of a singularity and its detrimental impact on the performance of planar flow remains unclear to readers. The paper asserts that the proposed approach enhances the stability of model training, setting the expectation for a detailed analysis.
> >
> > **A3** Thank you for your thorough comment. We should have explained more about the singularity issue. While we don't have a formal theoretical proof, we believe our approach is logically grounded. We would like to highlight the logical coherence in our method:
> >
> > The training instability and suboptimal convergence arising from the singularity are predominantly numerical issues. The presence of an unstable gradient around the singularity renders certain distributions within the variational family practically unattainable. Consequently, this limitation restricts the expressiveness of planar flows. By eliminating the singularity, we can explore all distributions during training, which leads to better approximation in general. We have incorporated this argument in the updated version.
> >
> >
> > > **Comment 2**: To meet this expectation, the paper should ideally include a comparative analysis of the gradient variance in planar flow, employing both the original and the new parameterization schemes. Additionally, a theoretical examination of the training dynamics in a deep planar flow setup would contribute to a more comprehensive understanding. The paper should also offer a conclusive answer to whether issues such as gradient explosion or vanishing are effectively circumvented with the proposed modifications. Without these elements, the paper's exploration of the singularity issue feels incomplete, leaving readers with unanswered questions and a lack of clarity on the significance of the proposed solution.
> >
> > **A4**: We trace the gradients of both flows during the training. Unfortunately, we did not find enough robust observations that can be included as a solid evidence. The gradients behavior of the old planar flow is not necessarily noise in all cases. We suspect that some regions of the variational family are not explored during training due to the singularity, thereby limiting the overall quality of the approximation.
> >
> > We made every effort to design a control experiment for comparing the original and the new planar flows, where all confounding factors are appropriately controlled (e.g., the initial parameters and training data). Hence, the only difference between the two flows is the reparameterization. We hope that our control experiments, conducted with a large number of replicates, can convincingly demonstrate that the improvement is indeed the result of eliminating the singularity.

---

> > > ### Comment · Reviewer_ndCL · 2023-11-21
> > > **Thank you for your response**
> > >
> > > I appreciate authors' response to my comments/questions and efforts for including additional experiments.  The additional comparison to NSF/sylvester flow does help to strengthen the usefulness of the method, because of which I'm willing to raise the score to 5. One caveat of this particular experiments conducted in Fig6 is that typically complicated flows such as NSF does not require large number of layers (for simple MNIST example, 10--20 layers would be more reasonable; the decaying performance of NSF with increasing flow length also confirms this), hence the comparison might be more insightful by also evaluating these flows at 5, 10, 20 layers.
> > >
> > > However, I still think issues noted in my previous comments deserves a more careful investigation.
> > >
> > > - to A3 and A4:  The argument---"The training instability and suboptimal convergence arising from the singularity are predominantly numerical issues. The presence of an unstable gradient around the singularity renders certain distributions within the variational family practically unattainable."---is intuitively correct but not grounded by any empirical evidence (as the author mentioned in A4---"we did not find enough robust observations that can be included as a solid evidence") or theoretical analysis. I personally appreciate the simple practical solution offered in this work, but rigorously analyzing the identified problem and developing in-depth understanding to the general issue leads to more significant impact.
> > >
> > > - to  A2: I understand that "planar flow does not have a closed form, making it less attractive for density estimation and data generation", but numerical inverse can be computed via root finders and works reliably in density estimation (see a julia implementation: https://github.com/TuringLang/Bijectors.jl/blob/04b79dd46eca8cea2f988348c47bd5e720a2b9a4/src/bijectors/planar_layer.jl#L112C1-L127C4).
> > > Because the scope of this work is relatively small, it worth analyzing the usefulness of the proposed reparameterization in all aspects.

---

> > > > ### Author Response · Authors · 2023-11-21
> > > >
> > > > Thank you sincerely for acknowledging the usefulness of our method. Your recognition of our efforts is truly appreciated. We reply to all your comments below.
> > > >
> > > >
> > > > > **Comment 1** One caveat of this particular experiments conducted in Fig6 is that typically complicated flows such as NSF does not require large number of layers (for simple MNIST example, 10--20 layers would be more reasonable; the decaying performance of NSF with increasing flow length also confirms this), hence the comparison might be more insightful by also evaluating these flows at 5, 10, 20 layers.
> > > >
> > > > **A5**: Indeed, the NSF does not require a large number of flow layers in practice. However, if a small number of flow layer is used for the NSF, then the neural network within each flow layer needs to be much larger and deeper. Since the planar flow has a fixed and small number of parameter per layer, for comparison purpose, we decrease the size of the neural network within the flow layer and increase the number of flow layers.
> > > >
> > > > Our intention was not to show that the singularity-free planar flows can outperform complicated flows in handling large image datasets with high dimension. The VAE experiment is one of the three tasks we conducted. We are hoping to develop an efficient and lightweight tool to solve general variational problems, possibly even on a low-configuration computer.
> > > >
> > > >
> > > > > **Comment 2** The argument---"The training instability and suboptimal convergence arising from the singularity are predominantly numerical issues. The presence of an unstable gradient around the singularity renders certain distributions within the variational family practically unattainable."---is intuitively correct but not grounded by any empirical evidence (as the author mentioned in A4---"we did not find enough robust observations that can be included as a solid evidence") or theoretical analysis. I personally appreciate the simple practical solution offered in this work, but rigorously analyzing the identified problem and developing in-depth understanding to the general issue leads to more significant impact.
> > > >
> > > > **A6**: We did find some empirical evidence on the parameter gradients, but the observation is not robust enough to design a well-defined experiment. For example, the gradient under the singular reparameterization might suddenly become very large, causing the model parameters to boost to unexpectedly large values. This observation occurs randomly and to varying degree. We find it hard to quantify this random behavior for all parameters and gradients. However, this numerical issue is reflected robustly on the training loss and final converged state. Hence, we design various tasks to demonstrate this impact.
> > > >
> > > > We completely understand and appreciate your emphasis on rigorous analysis. While I agree that in-depth theoretical scrutiny can enhance the impact of a solution, I would like to stress that the issue at hand is simple and numerical. Further theoretical analysis might not help to improve the effectiveness of the planar flows.
> > > >
> > > >
> > > > > **Comment 3** to A2: I understand that "planar flow does not have a closed form, making it less attractive for density estimation and data generation", but numerical inverse can be computed via root finders and works reliably in density estimation (see a julia implementation: https://github.com/TuringLang/Bijectors.jl/blob/04b79dd46eca8cea2f988348c47bd5e720a2b9a4/src/bijectors/planar_layer.jl#L112C1-L127C4). Because the scope of this work is relatively small, it worth analyzing the usefulness of the proposed reparameterization in all aspects.
> > > >
> > > > **A7**: We understand that the inverse can be computed numerically and it is possible to use planar flows in the area of density estimation and data generation. While the planar flow would then lose its strength of simplicity, we agree that further development in this area can be considered for future work.

---

> > > > > ### Comment · Reviewer_ndCL · 2023-11-22
> > > > >
> > > > > Thank you for the ongoing discussion. However, I respectfully maintain my stance and have decided to retain the current score. Regardless, I wish you the best of luck with your submission.

---

> > > > > > ### Author Response · Authors · 2023-11-22
> > > > > >
> > > > > > Thank you for your continued engagement and consideration. We respect your decision to maintain the current score and sincerely appreciate your well-wishes.

---

### Official Review · Reviewer_seL3 · 2023-10-26

**Soundness:** 3 good
**Presentation:** 3 good
**Contribution:** 2 fair
**Rating:** 5
**Confidence:** 5

**Summary:**

The authors proposed a new parameterization for planar flows that removes the singularity. Experiments on a range of problems validate the effectiveness of the proposed methods.

**Strengths:**

The observation of a singularity in the original parameterization of planar flows is new and important. The authors have done a nice job that motivates the new parameterization and the writing is clear.

**Weaknesses:**

1. No theoretically justification why the new parameterization that removes the singularity would makes the approximation better.

2. The experiments are all in relatively low dimensional space (the VAE example uss a latent dimension D=20). It is not clear if the benefit of the proposed method would extend to higher dimension problems.

**Questions:**

1. It seems that SNF has similar issue in its parameterization. How did you implement it in your experiments?

2. Figure 6 shows an interesting result. First, it is an unfair comparison between planar flows and IAF in terms of number of layers since each layer in IAF is more complicated and requires more parameters, and this makes the performance of planar flows even better. Second, it seems that the more powerful IAF does not do well in terms of posterior estimation in this case. Is it because the training is not long enough? Was the KL reported in the right plot the average over all test images? Can the author provide more direct posterior comparison (e.g., scatter plot of samples) to a ground truth MCMC run?

3. It seems that the advantage of the new parameterization decreases as the number layers increase, any explanations? Also, for planar flows, a large number of layers are often used in practice which makes the benefit of new parameterization a bit less attractive.

---

> ### Author Response · Authors · 2023-11-21
>
> # Response to Reviewer seL3
>
> Thank you for your thorough review and positive feedback. Your acknowledgment of the importance and the effectiveness of our proposed methods is encouraging.
>
> We reply to all your comments below.
>
>
> > **Weakness 1**: No theoretically justification why the new parameterization that removes the singularity would makes the approximation better.
>
> **A1**: We understand your concern. While we don't have a formal theoretical proof, we believe our approach is logically grounded. We would like to highlight the logical coherence in our method, which, while not a formal theoretical proof, serves as a rational basis for the approximation improvement.
>
> The training instability and suboptimal convergence arising from the singularity are predominantly numerical issues. The presence of an unstable gradient around the singularity renders certain distributions within the variational family practically unattainable. Consequently, this limitation restricts the expressiveness of planar flows. By eliminating the singularity, we can explore all distributions during training, which leads to better approximation in general.
>
>
> > **Weakness 2**: The experiments are all in relatively low dimensional space (the VAE example uss a latent dimension D=20). It is not clear if the benefit of the proposed method would extend to higher dimension problems.
>
> **A2**: We appreciate your insight into the potential limitations of low-dimensional experiments. We took your suggestion and run the VAE experiment with a higher dimension $D=40$. Similar results are observed. This consistency supports the robustness of our findings, suggesting that the observations are not merely circumstantial but hold in higher-dimensional spaces as well. We have included this experiment in Appendix G.
>
> Note that we did not run the IAF and NSF for $D=40$ as changing to a higher dimension necessitates a corresponding adjustment in the neural network size of IAF and NSF. Unfortunately, we did not have sufficient time to thoroughly test the neural network size to ensure comparable performance.

---

> > ### Author Response · Authors · 2023-11-21
> >
> > > **Question 1**: It seems that SNF has similar issue in its parameterization. How did you implement it in your experiments?
> >
> > **A3**: Yes, the SNF has similar constraints on its parameters, specifically, the diagonal entries of the two triangular matrices: $r_{ii} \tilde{r}_{ii} > -1$. We followed the implementation by its proposers, van den Berg et al. (2018) — applying $tanh()$ to all diagonal entries. We note that it is possible to adapt our patch for the Sylvester flow to improve its expressivity/trainability.
> >
> >
> > > **Question 2**: Figure 6 shows an interesting result. First, it is an unfair comparison between planar flows and IAF in terms of number of layers since each layer in IAF is more complicated and requires more parameters, and this makes the performance of planar flows even better. Second, it seems that the more powerful IAF does not do well in terms of posterior estimation in this case. Is it because the training is not long enough? Was the KL reported in the right plot the average over all test images? Can the author provide more direct posterior comparison (e.g., scatter plot of samples) to a ground truth MCMC run?
> >
> > **A4**: Thank you for raising this question. We should have explained more on the results of IAFs. First, we answer the concern about the training and metrics reported.
> >   - The training is long enough for all flows.
> >   - The KL reported in Figure 6 is the difference between $\log p(x)$ and ELBO, which are averaged over all 10,000 test images. For each test images, we estimate with 500 importance samples. We made every effort to ensure the accuracy of the metrics we reported.
> >   - Unfortunately, we are not able to provide a more direct posterior comparison to a ground truth MCMC run since the true posteriors are unknown.
> >
> > We have conducted an extra experiment with neural spline flows (NSFs). The updated comparison reaffirms the parameter efficiency of the new planar flows.
> >
> > *Reason why IAFs and NSFs do not perform well in our study*: While the IAFs and NSFs employed in this study have much larger numbers of parameters compared to the planar flows, it is essential to note that their sizes remain relatively small in the context of the neural network architecture. Also, training of such intricate models with deep layers may require extra attention. Consequently, the results do not reflect their optimal performance and are solely used for comparison in terms of parameter efficiency. We have incorporated this comment in the revised version (page 9).
> >
> >
> > **Question 3**: It seems that the advantage of the new parameterization decreases as the number layers increase, any explanations? Also, for planar flows, a large number of layers are often used in practice which makes the benefit of new parameterization a bit less attractive.
> >
> > **A5**: The advantage of the new parameterization does not always decrease as the number layers increases. For example, Figure 6 shows that the improvement is larger as the number of layers increases. In general, the expressiveness of the planar flow increases as the number of layers increases, which might cover the issue arose from the singularity. However, the unstableness makes it all unpredictable. This numerical issue might be more severe as the number of layers increases. Hence, we suggest to use the singularity-free planar flow to avoid any undesirable complications.

---

> > > ### Comment · Reviewer_seL3 · 2023-11-22
> > >
> > > Thanks for your response. I am still not sure the empirical findings are enough to fully convince the claim and hence would like to keep my score.

---

> > > > ### Author Response · Authors · 2023-11-22
> > > >
> > > > Thank you for getting back to us. We understand and respect your decision to maintain your current score.

---

### Official Review · Reviewer_KdQZ · 2023-10-30

**Soundness:** 3 good
**Presentation:** 3 good
**Contribution:** 3 good
**Rating:** 6
**Confidence:** 3

**Summary:**

This paper identifies a non-removable singularity in the original planar flow's parameterization, and proposes an new parameterization that removes this singularity and stabilizes the flow training. Empirically, they have shown this new parameterization leads to faster training and superior performance against the original parameterization on several synthetic experiments. In addition, they apply their new planar flow to VAE training on MNIST and find that the planar flow with the proposed parameterization achieves competitive performance against more advanced flow methods like IAF and SNF.

**Strengths:**

- This method is simple and plausible. Despite its simplicity, the experiments in the paper do demonstrate the improved performance of the planar flow. Although it is not clear whether this method is comparable to the latest flow methods on density estimation, this method could be useful in certain variational inference problems where a simple flow model is needed.
- The paper is well-written and easy to understand.

**Weaknesses:**

- The particular choice of the $m(\cdot)$ function seems a bit arbitrary. As argued in the paper, all we need is $m(x) = \mathcal{O}(x)$ when $x \to 0$. There are many functions that satisfy this condition. To confirm that the issue of the original parameterization is indeed the singularity, the authors should pick a few other $m(\cdot)$ functions satisfying $m(x) = \mathcal{O}(x)$ and check if they share similar performance.
- The idea is so simple that it is merely a parameterization trick. I am not sure if something similar has been done before. However, this may not be an actual weakness of this paper, as the reviewer is not particularly familiar with the literature on the planar flow.
- One advantage of the planar flow in this paper is that its performance is comparable to the IAFs and SNFs but the planar flow has fewer parameters. One experiment to further strengthen this point is to compare the wall-clock running time of each flow.

**Questions:**

- Have the authors generated images from the trained VAEs on the MNIST as a sanity check?
- Have the authors tried other function forms of $m(\cdot)$ that satisfy the condition $m(x) = \mathcal{O}(x)$?

---

> ### Author Response · Authors · 2023-11-21
>
> # Response to Reviewer KdQZ
>
> Thank you for your thorough and insightful comments. We are happy to see that you find our method useful and our paper well-written.
>
> We reply to all the points below.
>
>
> > **Weakness 1**: The particular choice of the $m(\cdot)$ function seems a bit arbitrary. As argued in the paper, all we need is when $m(x) = \mathcal{O}(x)$ when $x \rightarrow 0$. There are many functions that satisfy this condition. To confirm that the issue of the original parameterization is indeed the singularity, the authors should pick a few other $m(\cdot)$ functions satisfying $m(x) = \mathcal{O}(x)$ and check if they share similar performance.
> >
> > **Question 2**: Have the authors tried other function forms of $m(\cdot)$ that satisfy the condition $m(x) = \mathcal{O}(x)$?
>
> **A1**: We regret that we did not make it clear how we chose the function $m(\cdot)$. The sufficient condition we stated was in the context of removing the singularity. There are other implicit conditions as well. The main role of $m(\cdot)$ is to squeeze the unconstrained dot product to the required region $(-1, \infty)$, which in turn reparameterizes the unconstrained $v'$ to the feasible values.
>
> We chose $m(x)=x$ if $x \ge 0$ and a simple continuously differentiable extension to $x<0$, so that a minimal reparameterization is imposed on the unconstrained $v'$. This choice aligns with our belief in simplicity and minimizes the computation required.
>
> Certainly, there are other more complicate functions satisfying all the conditions. However, they might just introduce unnecessary computation. We appreciate the reviewer for raising this question. We have modified the corresponding section to enhance clarity in the updated version (page 5).
>
>
> > **Weakness 2**: The idea is so simple that it is merely a parameterization trick. I am not sure if something similar has been done before. However, this may not be an actual weakness of this paper, as the reviewer is not particularly familiar with the literature on the planar flow.
>
> **A2**: We agree that it is a parameterization trick. To the best of our knowledge, our work is the first attempt to address this specific issue. Frankly, this work is not groundbreaking. Our objective is to provide a clear and accessible solution to general variational problems, particularly in scenarios where ease of implementation and computational efficiency are crucial.
>
>
> > **Weakness 3**: One advantage of the planar flow in this paper is that its performance is comparable to the IAFs and SNFs but the planar flow has fewer parameters. One experiment to further strengthen this point is to compare the wall-clock running time of each flow.
>
> **A3**: Thank you for your valuable suggestion. The planar flow is indeed faster and requires less computation resource than other flows due to its simplicity, especially with the minimal reparameterization we proposed. Based on our experiments, the new planar flows are approximately faster by a few hours than IAFs and SNFs for completing a replicate of the VAE experiment. We did not report this in the paper since the wall-clock running time we collected is not robust enough due to factors beyond our control.
>
> We also note that the wall-clock running time is not the only metric and does not fully demonstrate the strength of the planar flow. For example, the planar flow is also trained on GPUs (to facilitate a fair control experiment), while it only requires small GPU usage, hence low power consumption. Additionally, it is possible to train planar flows efficiently on CPU-only devices.
>
>
> > **Question 1**: Have the authors generated images from the trained VAEs on the MNIST as a sanity check?
>
> **A4**: Thank you for pointing it out. We have performed the sanity check as you suggested. The images generated from the trained VAEs are reasonable, which indicates that our models are properly trained. We have included the generated images in Appendix F.

---

> > ### Comment · Reviewer_KdQZ · 2023-11-21
> >
> > Hi authors,
> >
> > Thanks for clarifications. I think those comments make sense.
> >
> > The generated images on MNIST show that (a) the model is properly trained (b) the sample quality of the new parameterization is marginally better than the old parameterization.
> >
> > I think the value of this paper is a simple and computationally cheap flow model which could be beneficial in certain settings -- a particular example is variational inference. I still lean towards acceptance. However, since I myself is not an expert in flow based model, I am less willing to increase the score further. I think the authors' priority is to convince the other reviewers.

---

> > > ### Author Response · Authors · 2023-11-21
> > >
> > > Your recommendation for acceptance, coupled with your recognition of our simple and computationally efficient flow method, is truly encouraging.
> > >
> > > As for the score, we understand and respect your judgment and perspective.

---

### Official Review · Reviewer_Z1xs · 2023-11-01

**Soundness:** 3 good
**Presentation:** 3 good
**Contribution:** 2 fair
**Rating:** 6
**Confidence:** 5

**Summary:**

The authors consider the original formulation of normalizing flows parameterized by neural networks presented in the work by Rezende et al. They note that this parameterization has a deficiency (the function can become singular) that limits its expressivity/trainability.

They propose a patch to it and test it on standard benchmarks -- density estimation for challenging 2d densities, a Bayesian inference task, and the variational autoencoder formulation of the VI problem where an equivalent Evidence Lower Bound (ELBO) can be optimized.

**Strengths:**

I am a believer that a paper does not need to be groundbreaking to be a meaningful contribution to the literature. This paper does not try to be groundbreaking or to sell itself as such. It notes a deficiency in a foundational method, and patches that deficiency, and it justifies it with experiments.

Those experiments are thorough -- and for once it seems like the comparison they make to other methods is reproducible (often papers will make numerical comparisons to other works, claiming such and such number is better than such and such other number, but the experimental conditions leading to those discrepancies in numbers is not clearly causally related to the proposed method). While the experiments are simple and low dimensional, this can actually be a nice feature for a paper to suggest that things are indeed reproducible in a reasonable sense.

**Weaknesses:**

The other perspective on the simplicity of the proposal is that it is not necessarily quite intellectually rich. I say this in the following sense - the related works points to many papers that choose different parameterizations for the coupling functions in normalizing flows. A pushback then is to say that changing a parameterization in the form they've done here is not that conceptually different from just proposing other transformations that don't have the original issue in the planar flows. The authors make a remark about the parameter efficiency of this method (which is true, it's a very simple function), but hopefully there'd be a bit more here.

Of particular relevance is the Sylvester Flows paper, which presents itself as a generalization of the planar flows that this paper is addressing.

Some related work that the authors should include:

**Other works that lay the groundwork for flow-based variational inference in scientific computing:**
- Boltzman Generators, *Frank Noé, Simon Olsson, Jonas Köhler, Hao Wu*, 2019
- Flow-based generative models for Markov chain Monte Carlo in lattice field theory, *Michaels S. Albergo, Gurtej Kanwar, Phiala E. Shanahan*, 2019.

**Questions:**

- Can the authors explain if their patch is relevant to the generalization of planar flows (Sylvester flows)?
- Can the authors justify their claim of parameter efficiency by providing parameter counts and pushing e.g. the number of parameters in the neural networks in the neural spline flows down to their minimum to maintain performance? Splines are for example clearly more expressive, the question is just what is the minimal network size that makes them competitive with the proposal here. This doesn't seem like that hard of an experiment to run either, but of course I am sympathetic to the overhead of more experimentation.
- Are there any higher dimensional experiments that the authors could run? The downside of low-d is it's hard to understand if the observations are generic or circumstantial.

---

> ### Author Response · Authors · 2023-11-21
>
> # Response to Reviewer Z1xs
>
> We sincerely appreciate your progressive perspective on the nature of meaningful contributions to the literature. Your recognition that a paper doesn't necessarily need to be groundbreaking to make a significant impact resonates deeply with our intentions behind this work.
>
> We reply to all your comment, suggestion, and questions below.
>
> > **Weakness 1**: The other perspective on the simplicity of the proposal is that it is not necessarily quite intellectually rich. I say this in the following sense - the related works points to many papers that choose different parameterizations for the coupling functions in normalizing flows. A pushback then is to say that changing a parameterization in the form they've done here is not that conceptually different from just proposing other transformations that don't have the original issue in the planar flows. The authors make a remark about the parameter efficiency of this method (which is true, it's a very simple function), but hopefully there'd be a bit more here.
>
> **A1**: We acknowledge that our approach is intentionally simple, and we understand the concern about its conceptual depth. In choosing simplicity, our intention was to provide a clear and accessible solution to general variational problems, particularly in scenarios where ease of implementation and computational efficiency are crucial.
>
>
> > **Suggestion 1**: Some related work that the authors should include:
> > Other works that lay the groundwork for flow-based variational inference in scientific computing:
> >   - Boltzman Generators, Frank Noé, Simon Olsson, Jonas Köhler, Hao Wu, 2019
> >   - Flow-based generative models for Markov chain Monte Carlo in lattice field theory, Michaels S. Albergo, Gurtej Kanwar, Phiala E. Shanahan, 2019.
>
> **A2**: Thank you for your valuable suggestion. We have diligently reviewed the papers you recommended, and incorporated references to these works in the updated version (page 2).
>
>
> > **Question 1**: Can the authors explain if their patch is relevant to the generalization of planar flows (Sylvester flows)?
>
> **A3**: The Sylvester flow is considered as the generalization of the planar flow since it has the same form, $f(z) = z + A h(Bz + b)$, but allows for matrix weights. Our patch to the planar flow does not change the dimension of the weight parameters, so it is different from the approach of the Sylvester flow. However, there are similar constraints on the parameters of the Sylvester flow, $r_{ii} \tilde{r}_{ii} > -1$. Hence, it is possible to adapt our patch for the Sylvester flow to improve its expressivity/trainability.
>
>
> > **Question 2**: Can the authors justify their claim of parameter efficiency by providing parameter counts and pushing e.g. the number of parameters in the neural networks in the neural spline flows down to their minimum to maintain performance? Splines are for example clearly more expressive, the question is just what is the minimal network size that makes them competitive with the proposal here. This doesn't seem like that hard of an experiment to run either, but of course I am sympathetic to the overhead of more experimentation.
>
> **A4**: Thank you for your suggestion regarding the experimental design for neural spline flows (NSFs). Your experimental design for NSFs aligns with our treatment to IAFs. We promptly conducted this experiment; however, it's worth noting that the training for NSFs required a considerably longer duration compared to other flow methods. We apologize for the delayed response.
>
> The updated comparison reaffirms the parameter efficiency of the new planar flows. We have included the results of the NSFs experiment in Figure 6, and a table summarizing the parameter counts is now available in Appendix D in the updated version.
>
>
> > **Question 3**: Are there any higher dimensional experiments that the authors could run? The downside of low-d is it's hard to understand if the observations are generic or circumstantial.
>
> **A5**: We appreciate your insight into the potential limitations of low-dimensional experiments. We took your suggestion and run the VAE experiment with a higher dimension $D=40$. Similar results are observed. This consistency supports the robustness of our findings, suggesting that the observations are not merely circumstantial but hold in higher-dimensional spaces as well. We have included this experiment in Appendix G.
>
> Note that we did not run the IAF and NSF for $D=40$ as changing to a higher dimension necessitates a corresponding adjustment in the neural network size of IAF and NSF. Unfortunately, we did not have sufficient time to thoroughly test the neural network size to ensure comparable performance.

---

> > ### Comment · Reviewer_Z1xs · 2023-11-22
> > **Thanks for your improvements**
> >
> > Thanks for your improvements, I have raised my score accordingly to be over the threshold. It is an acceptable set of experiments.

---

> > > ### Author Response · Authors · 2023-11-22
> > >
> > > We sincerely appreciate your recognition and are grateful for the opportunity to address your concerns. Your positive reevaluation means a lot to us.

---

### Meta-Review · Area_Chair_KoNi · 2023-12-06

**Metareview:**

This paper revisited the planar flows and identified a non-removable singularity in the original formulation. To address this the paper proposed a new parameterisation to remove this singularity. Experiments on several synthetic datasets and VAE training on MNIST demonstrated improvements over the old version planar flows and made them competitive when compared with more advanced flow methods such as IAF and SNF.

While reviewers all agree that "reviving" previous techniques by finding and removing its undiscovered pitfalls is a valuable contribution, they are not satisfied with the empirical evidence provided by the paper to support the proposed methodology. In particular, they asked about density estimation experiments as well as stronger baselines for the Bayesian regression experiment (by e.g., using HMC).

**Justification For Why Not Higher Score:**

The idea can be useful but at the current stage the empirical evidence is not solid enough to support the proposed method.

**Justification For Why Not Lower Score:**

N/A

---

### Decision · Program_Chairs · 2024-01-16

Reject